# Recommendations for hand hygiene in community settings: a scoping review of current international guidelines

Clara MacLeod ,[1] Laura Braun,[1] Bethany A Caruso,[2] Claire Chase,[3] Kondwani Chidziwisano,[4] Jenala Chipungu,[5] Robert Dreibelbis,[1] Regina Ejemot-Nwadiaro,[6] Bruce Gordon,[7] Joanna Esteves Mills,[7] Oliver Cumming [1]

For numbered affiliations see end of article.

**Correspondence to**
Clara MacLeod;
clara.macleod@lshtm.ac.uk

## ABSTRACT

**Background** Hand hygiene is an important measure to prevent disease transmission.

**Objective** To summarise current international guideline recommendations for hand hygiene in community settings and to assess to what extent they are consistent and evidence based.

**Eligibility criteria** We included international guidelines with one or more recommendations on hand hygiene in community settings—categorised as domestic, public or institutional—published by international organisations, in English or French, between 1 January 1990 and 15 November 2021.

**Data sources** To identify relevant guidelines, we searched the WHO Institutional Repository for Information Sharing Database, Google, websites of international organisations, and contacted expert organisations and individuals.

**Charting methods** Recommendations were mapped to four areas related to hand hygiene: (1) effective hand hygiene; (2) minimum requirements; (3) behaviour change and (4) government measures. Recommendations were assessed for consistency, concordance and whether supported by evidence.

**Results** We identified 51 guidelines containing 923 recommendations published between 1999 and 2021 by multilateral agencies and international non-governmental organisations. Handwashing with soap is consistently recommended as the preferred method for hand hygiene across all community settings. Most guidelines specifically recommend handwashing with plain soap and running water for at least 20 s; single-use paper towels for hand drying; and alcohol-based hand rub (ABHR) as a complement or alternative to handwashing. There are inconsistent and discordant recommendations for water quality for handwashing, affordable and effective alternatives to soap and ABHR, and the design of handwashing stations. There are gaps in recommendations on soap and water quantity, behaviour change approaches and government measures required for effective hand hygiene. Less than 10% of recommendations are supported by any cited evidence.

**Conclusion** While current international guidelines consistently recommend handwashing with soap across community settings, there remain gaps in recommendations where clear evidence-based guidance might support more effective policy and investment.

## STRENGTHS AND LIMITATIONS OF THIS STUDY

⇒ The scoping review follows the Arksey and O'Malley methodological framework to identify, retrieve and summarise current international guideline recommendations for hand hygiene in community settings.

⇒ Various sources were systematically searched to identify relevant guidelines published by multilateral agencies and international non-governmental organisations between January 1990 and November 2021.

⇒ The search was limited to guidelines published in English or French and therefore may have excluded relevant guidelines published in other languages.

⇒ The quality of the included guidelines was not assessed, although we did consider the extent to which recommendations were based on cited evidence.

## INTRODUCTION

Hand hygiene, including handwashing with soap and other methods such as alcohol-based hand rubs (ABHRs), is an important public health measure that can prevent the transmission of a range of diseases.[1] Handwashing with soap has been found to be a cost-effective intervention[2] that can reduce the risk of both diarrhoeal disease and acute respiratory infections by over 20%.[3–9] Handwashing with soap has also been linked to the reduction of certain neglected tropical diseases, including trachoma and some soil-transmitted helminth infections.[10 11] Recently, handwashing with soap and the use of ABHRs were advised as one of the key control measures during the COVID-19 pandemic[12 13] and were found to be effective.[14]

This scoping review focuses on hand hygiene in non-healthcare settings, which we collectively refer to as 'community settings'. Using the definition set out in the Ottawa Charter, we consider settings as where 'health is created and lived by people within the settings of their everyday life; where they

learn, work, play and love',[15] and include domestic, public and institutional settings. The practice of hand hygiene—and access to the facilities which enable this—is often limited in these community settings, particularly in low/middle-income countries.[16] In the domestic setting, 30% of the global population does not have access to a basic handwashing facility with soap and water at home,[17 18] with three-quarters of those living in low-income countries.[17] In institutional settings, an estimated 43% of schools worldwide do not have access to basic hand hygiene facilities,[18] but there are limited data for other institutional settings, such as the workplace and prisons and places of detention, and public settings, such as markets, transportation hubs, and places of worship.[18]

Despite the international recognition of hand hygiene as a critical public health measure, a recent global assessment of government policies, planning and financing for hygiene found that while the majority of surveyed countries reported having national policies for hand hygiene, less than 10% had sufficient financing to implement them.[19] Various international guidelines with recommendations on hand hygiene for non-healthcare settings exist,[13 20–22] but it is unclear whether current international guidelines are comprehensive, consistent and based on the most rigorous evidence available. This review aims to summarise current international recommendations for hand hygiene in community settings, identify areas of consistency and concordance, and assess whether recommendations are evidence based.

## METHODS
This review follows the six stages of the Arksey and O'Malley methodological framework for scoping reviews.[23–25] Expert consultation (stage 6) was integrated throughout the scoping review process[24] to obtain feedback on the scope and conceptual framework and to identify any additional guidelines beyond those identified during the electronic search. Our review is described according to the Preferred Reporting Items for Systematic reviews and Meta-Analyses extension for Scoping Reviews (PRISMA-ScR)[26] and a PRISMA-ScR checklist is included in the online supplemental table A1. The protocol was preregistered with OSF Registries.[27]

### Identifying the research question (stage 1)
To identify and refine the research question, a conceptual framework was developed for this review and built around three key concepts: (1) hand hygiene, (2) non-healthcare settings and (3) current international guidelines (figure 1). In this review, we define hand hygiene as any action of hand cleansing for the purpose of removing or deactivating pathogens from hands.[28] Effective hand hygiene is defined as any practice which removes or deactivates pathogens from hands and thereby limits disease transmission.[28]

The first key concept in the conceptual framework—hand hygiene—covers four areas: effective hand hygiene, minimum requirements, behaviour change and government measures (figure 1). Effective hand hygiene refers

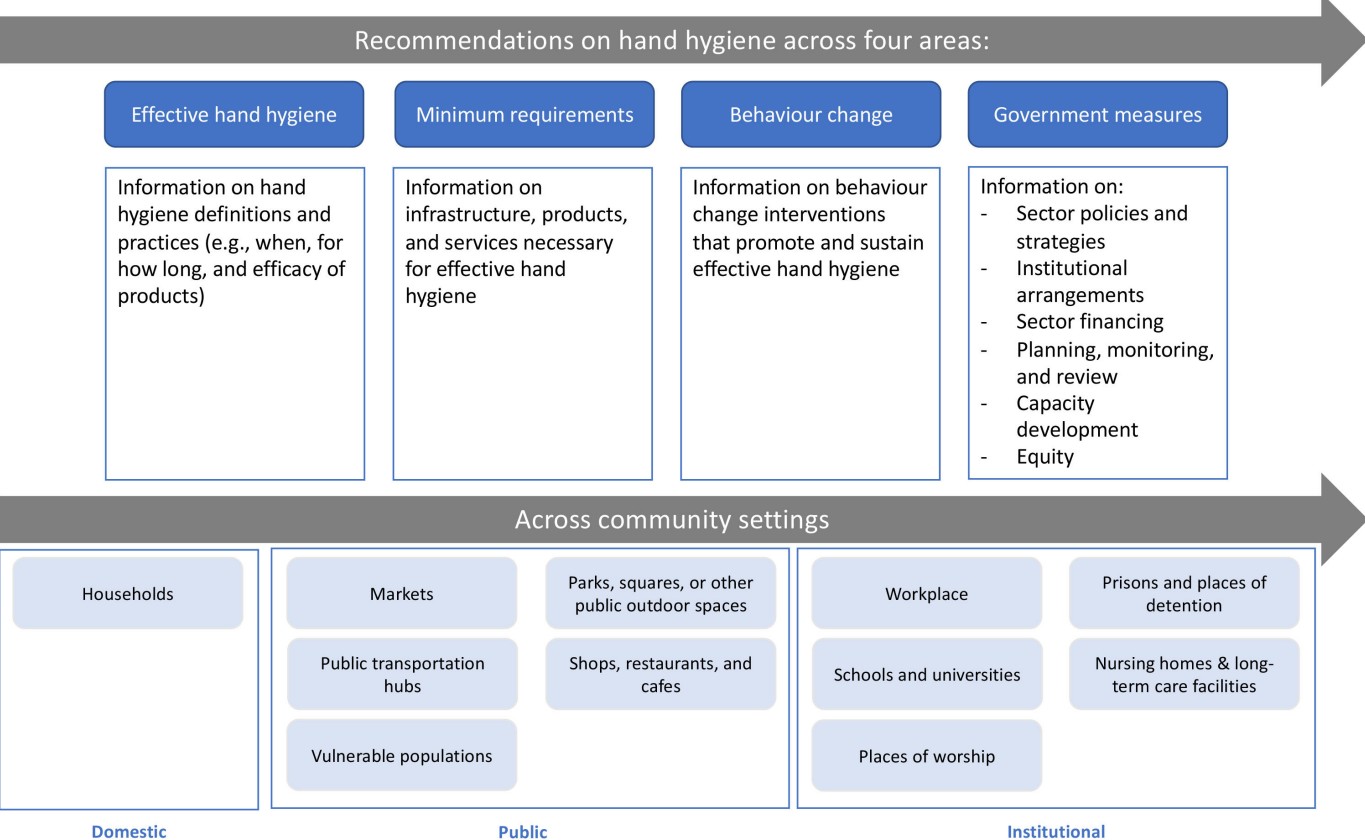

**Figure 1** Conceptual framework for hand hygiene in community settings.

to definitions and practices. Minimum requirements refer to the materials, services and infrastructure required for effective hand hygiene. Behaviour change denotes the appropriate behavioural performances that promote and sustain effective hand hygiene. Government measures concern actions taken by governments to ensure effective hand hygiene, which are categorised according to an established framework[29] as follows: policy and strategy; institutional arrangements; sector financing; planning, monitoring and review; capacity development; and equity.

The second key concept—community settings—is defined as settings where healthcare is not routinely delivered,[28] broadly spanning all places where people 'learn, play, work and love'[15] and specifically including domestic, public and institutional settings. As these settings may not exclusively refer to physical settings,[30] the review also includes recommendations for vulnerable populations (eg, people experiencing homelessness) who may reside permanently or semipermanently in public spaces. Nursing homes, long-term care facilities, non-acute care facilities and home care are also included in the review under institutional settings, as the boundary between healthcare and non-healthcare settings is often unclear or recommendations for these subsettings may not be disaggregated, especially in the context of COVID-19 where people received care at home.

The third concept—guideline—is defined as a published document where the primary purpose is to provide specific guidance, in the form of recommendations, towards a course of action. A recommendation is defined as a statement designed to assist a targeted actor to make an informed decision on whether, when and how to undertake a specific action.[31] Our review is limited to international guidelines to identify generalisable recommendations of global relevance.

### Identifying relevant studies (stage 2)

To identify relevant guidelines, the search strategy consisted of the following sources: (1) the WHO Institutional Repository for Information Sharing (IRIS) Database, (2) Google search engine, (3) websites of international organisations known to work on hand hygiene (online supplemental file 1) and (4) contacting experts. We searched the WHO IRIS Database using prespecified search terms related to hand hygiene, non-healthcare settings and guidelines (online supplemental table A3). The search in Google was carried out using the anonymous function in the web browser (Chrome) to reduce the influence of the reviewer's (CM) individual search history. Search strings were constructed by using multiple combinations of search terms from online supplemental table A3. The first 10 pages of Google were reviewed by one reviewer (CM). Documents that at first appeared related to the research question and met the inclusion criteria were included for further screening. Expert consultations were conducted with 'Hand Hygiene for All' Initiative core partners (online supplemental table A4) to identify potentially relevant guidelines. The reference lists

of guidelines were also hand-searched for any additional relevant documents. The search was limited to English and French languages and publication date was restricted to 1 January 1990 onwards to identify current guidelines.

### Study selection (stage 3)

Documents meeting the following criteria were included: (1) international guideline, (2) offers one or more recommendations on hand hygiene, (3) targets at least one community setting, as defined in the conceptual framework, (4) published by an international non-governmental organisation (NGO), multilateral agency or public health agency, (5) published in English or French, and (6) published between 1 January 1990 and 15 November 2021. The review excludes guidelines for humanitarian settings, as internationally agreed guidance on hand hygiene in humanitarian settings and complex emergencies is available through the Sphere standards for water, sanitation and hygiene (WASH) promotion.[32] Only the most recent versions of guidelines were included, with previous versions of the same guidelines excluded. Country-specific guidelines were also excluded.

All documents retrieved from electronic searches and expert consultations were transferred to Mendeley[33] for de-duplication. Inclusion was completed in two stages: (1) title and abstracts were screened for eligibility by one reviewer (CM); and (2) full texts of all potentially eligible documents were retrieved and independently assessed for inclusion by two reviewers (CM and LB). Disagreement between reviewers on inclusion was resolved through arbitration by a third reviewer (OC).

### Charting the data (stage 4)

Guideline characteristics and recommendations from included guidelines were double-extracted by two reviewers (CM and LB) using a standardised data extraction template in MS Excel[34] and then cross-checked for accuracy. As with inclusion, a third reviewer (OC) provided arbitration if agreement on extraction could not be reached. The data extraction form (online supplemental table A5) included information on guideline characteristics, such as author, year of publication, target setting and COVID-19 response, as well as 57 specific parameters related to the four areas of hand hygiene described in the conceptual framework (figure 1). Recommendations for each parameter were extracted from included guidelines where possible.

### Collating, summarising and reporting the results (stage 5)

Guideline recommendations were first summarised for each parameter across community settings and then disaggregated by domestic, public or institutional setting where relevant. Definitions and recommendations for hand hygiene were assessed for consistency, concordance and whether supported by cited evidence. Recommendations were classified as consistent, fairly consistent or inconsistent if they featured in 10 or more, 4–9 or less than 4 guidelines, respectively.

Concordance is here defined as parameters with no consistent or fairly consistent recommendations at odds with each other. We also used a hierarchical system to classify evidence cited for recommendations into five levels, adapted according to an established classification system[35]: (1) systematic review or meta-analysis, (2) randomised controlled trial or controlled trial without randomisation (eg, quasi-experimental), (3) guideline (either developed from systematic reviews or not developed from systematic reviews), (4) observational study (eg, cohort, cross-sectional, case–control studies), and (5) expert opinion or anecdotal information (eg, programme documentation). A recommendation was considered evidence based if the guideline provided a specific citation for the recommendation, which was coded according to the level of evidence cited. Finally, evidence gaps were defined as parameters with very few recommendations (ie, less than 10 recommendations, equivalent to less than 20% of guidelines providing a recommendation).

## Patient and public involvement

There was no public or patient involvement in the course of this project.

## RESULTS

### Search results

Electronic searches were conducted on 15 November 2021, identifying 3360 records (2432 from the WHO IRIS Database, 900 from Google, 28 from international agency websites) (online supplemental table A2) and a further 11 records identified through expert consultation. Following de-duplication, a total of 3132 records were screened by title and abstract, and 125 documents were sought for retrieval for full-text screening, with one document not accessible. Finally, 51 guidelines are included in the review (figure 2). The 73 documents excluded during the full-text review are listed in the online supplemental table A6 with reasons for exclusion.

### Description of included guidelines

The 51 included guidelines were published in English between 1999 and 2021, with 38 published in 2020 or later and 31 providing guidance specifically on hand hygiene to help prevent the transmission of COVID-19. Among the 51 included guidelines, 67% are published by multilateral agencies (WHO, UNICEF, United Nations High Comissioner for Refugees and International Labour Organization), 23% by international NGOs, and 10% by the US Centers for Disease Control and Prevention

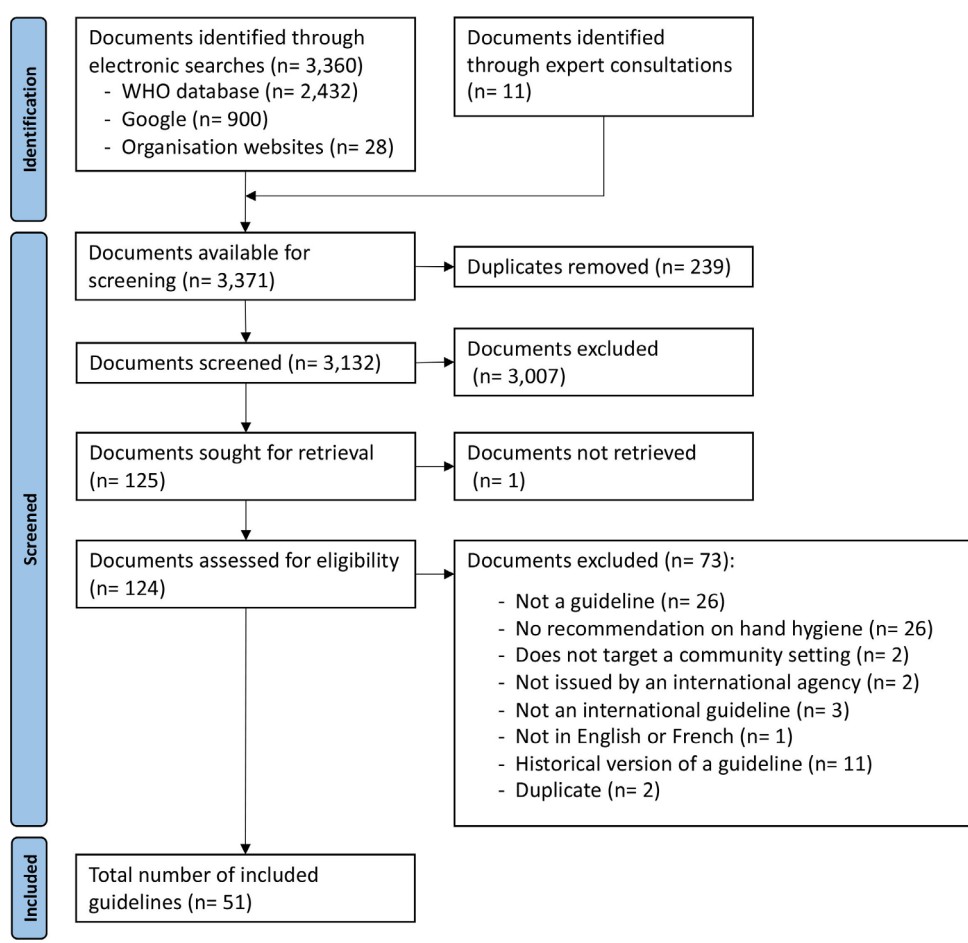

**Figure 2** Preferred Reporting Items for Systematic Reviews and Meta-Analyses flow diagram.

(online supplemental table A7). Most guidelines target public and institutional settings, while none exclusively target the domestic setting. More specifically, 43% (n=22) target the public setting, 43% (n=22) the institutional setting, 4% (n=2) the domestic and public setting, and 10% (n=5) more than one setting (eg, domestic, public and/or institutional). Of the 22 guidelines for the public setting, 20 concern public spaces and 2 concern vulnerable populations within public spaces (eg, people experiencing unsheltered homelessness and people living in dense, informal settlements). For institutional settings, eight guidelines concern schools, six the workplace, four prisons and places of detention, three places of worship, and one long-term care facilities and home care. Almost no guidelines, however, define a community setting. The majority of recommendations are generalisable to high-income, medium-income and low-income settings. Overall, we extracted 923 recommendations from the 51 included guidelines (online supplemental table A8).

### Recommendations for effective hand hygiene
#### Hand hygiene definitions
Only 10% of guidelines provide a clear definition for hand hygiene or safe or effective hand hygiene.[36–40] Meanwhile, 75% of guidelines provide at least one recommendation on when to practise hand hygiene, referred to in our review as a 'key moment'. Guidelines use inconsistent terms for defining when to practise hand hygiene, such as 'key times' (14%),[37 41–46] 'critical times' (8%)[47–50] and 'key moments' (4%).[21 51] Otherwise, guidelines either do not use a specific term for defining a key moment (49%) or do not recommend at least one key moment (25%).

#### Hand hygiene practices
There is agreement among guidelines on what constitutes effective hand hygiene across community settings, although there are gaps in recommendations on how to practise effective hand hygiene. Almost all guidelines (90%) recommend washing hands with soap and for the duration of at least 20 s (27%) (table 1). In addition, 63% of guidelines recommend the use of ABHR as an alternative or complement to handwashing with soap, though very few guidelines recommend duration for hand rubbing. There is a lack of recommendations on handwashing or hand rubbing technique (16%),[21 40–42 47 52–54] with two of these guidelines (25%, n=2)[21 40] referring to the WHO instructions for hand hygiene in healthcare settings.[28] Meanwhile, 24% of guidelines recommend the inclusion or provision of simple instructions on handwashing technique in hygiene promotion programmes, yet none of these guidelines include specific steps on handwashing or hand rubbing technique.

Guidelines provide inconsistent recommendations on when to practise hand hygiene. Over 30 different individual key moments are recommended among the guidelines. Despite this inconsistency, the individual key moments most commonly mentioned include 'before and after eating', 'after using the toilet', 'before and

after preparing food', 'after blowing nose, coughing or sneezing', and 'after touching public surfaces or objects' (table 1). The latter two feature most commonly among guidelines published during the COVID-19 pandemic. Fairly consistently recommended individual key moments include 'when entering or exiting the home or public space', 'after changing a child's diaper' and 'when hands are visibly dirty'. Almost all guidelines recommend clusters of key moments (ie, individual key moments recommended together). The most common ones are 'after using the toilet' with 'before and after eating' (39%, n=20); 'after using the toilet' with 'before and after preparing food' (24%, n=12); and 'after using the toilet' with 'before and after eating', and 'after blowing nose, coughing or sneezing' (22%, n=11).

### Recommendations for minimum requirements
#### Handwashing materials
Guidelines consistently recommend plain soap for handwashing across all community settings and paper towels or electric dryers for hand drying in public and institutional settings. Most guidelines (65%) do not recommend a specific type of soap for handwashing, though some specifically recommend bar soap (24%), liquid soap (22%) or 'soapy water' (eg, home-made mixture of powder or liquid soap, or bar soap shavings diluted in water) (16%) (table 1). No guidelines specifically recommend antibacterial soap. For hand drying after handwashing, most guidelines recommend clean, single-use paper towels (39%) and electric air-drying systems (22%) in public and institutional settings. Eight per cent of guidelines recommend a bin with disposal liners and lid for the waste management of paper towels in public and institutional settings.[21 55–57] No guidelines provide a recommendation for hand drying materials in the domestic setting.

#### Soap and water requirements for handwashing
While guidelines consistently recommend running water for handwashing with soap, there are gaps for soap and water quantity and discordant recommendations on water quality for handwashing. For water services, almost half (43%) of guidelines recommend running water, 16% a piped water system connected to a tap, and 14% water trucking, storage or manual refilling. Few guidelines recommend a minimum quantity of liquid or bar soap (14%)[43 48 58–61] or water (18%)[12 20 47 50 60–62] required for handwashing (eg, minimum amount of water needed to wet hands or soap product necessary to cover all surfaces of hands for handwashing with soap) (table 2). One guideline, for example, recommends less than 5 mL of liquid soap per person per handwashing event,[60] whereas three other guidelines recommend at least 250 g of bar soap per person per month for handwashing.[43 48 61] For water quality, 16% of guidelines state that water must be of drinking water quality, in line with WHO guidelines.[21 40 41 52 54 63–66] In contrast, 8% state that it does not have to be of drinking water quality,[12 20 39 67] though none

**Table 1** Consistent and fairly consistent recommendations for hand hygiene in community settings

| Parameter | Recommendation | All guidelines (n=51) | Domestic setting (n=2) | Public setting (n=24) | Institutional setting (n=22) | Multiple settings (n=5) |
|---|---|---|---|---|---|---|
| **Effective hand hygiene** | | | | | | |
| Key moments | Before and after eating | 45% (23) | 0% (0) | 38% (9) | 27% (14) | 0% (0) |
| | After using the toilet | 41% (21) | 0% (0) | 38% (9) | 55% (12) | 0% (0) |
| | Before and after preparing food | 27% (14) | 0% (0) | 38% (9) | 18% (4) | 20% (1) |
| | After blowing nose, coughing or sneezing | 24% (12) | 0% (0) | 21% (5) | 27% (6) | 20% (1) |
| | After touching public surfaces or objects | 24% (12) | 0% (0) | 12% (6) | 18% (4) | 40% (2) |
| | When entering and exiting buildings or home | 14% (7) | 0% (0) | 17% (4) | 5% (1) | 40% (2) |
| | After changing a child's diaper | 12% (6) | 0% (0) | 17% (4) | 0% (0) | 40% (2) |
| | When hands visibly dirty | 10% (5) | 0% (0) | 13% (3) | 9% (2) | 0% (0) |
| | Before and after work | 8% (4) | 0% (0) | 0% (0) | 18% (4) | 0% (0) |
| | After contact with animals | 8% (4) | 0% (0) | 13% (3) | 5% (1) | 0% (0) |
| | Taking care of sick person | 8% (4) | 0% (0) | 13% (3) | 5% (1) | 0% (0) |
| Handwashing duration | At least 20 s | 27% (14) | 0% (0) | 21% (5) | 36% (8) | 20% (1) |
| **Minimum requirements for effective hand hygiene** | | | | | | |
| **Materials for effective hand hygiene** | | | | | | |
| Soap | Soap | 65% (33) | 100% (2) | 50% (12) | 82% (18) | 20% (1) |
| | Bar soap | 24% (12) | 0% (0) | 14% (7) | 14% (3) | 40% (2) |
| | Liquid soap | 22% (11) | 0% (0) | 14% (7) | 14% (3) | 20% (1) |
| | Soap water | 16% (8) | 0% (0) | 17% (4) | 5% (1) | 60% (3) |
| ABHR | ABHR with at least 60% alcohol | 35% (18) | 0% (0) | 16% (8) | 45% (10) | 0% (0) |
| | ABHR, no alcohol percentage | 22% (11) | 50% (1) | 17% (4) | 23% (5) | 20% (1) |
| Hand drying | Clean, single-use paper towels | 39% (20) | 0% (0) | 14% (7) | 59% (13) | 0% (0) |
| | Air-drying system | 22% (11) | 0% (0) | 17% (4) | 14% (7) | 0% (0) |
| | Bin with disposable liners and lid | 8% (4) | 0% (0) | 4% (1) | 14% (3) | 0% (0) |
| **Alternative materials** | | | | | | |
| Other materials | Ash | 22% (11) | 0% (0) | 12% (6) | 14% (3) | 40% (2) |
| Conditional recommendations | ABHR where soap and water not available | 18% (9) | 0% (0) | 4% (1) | 36% (8) | 0% (0) |
| | Ash and water if ABHR or soap not available | 14% (7) | 0% (0) | 13% (3) | 14% (3) | 20% (1) |
| | ABHR if hands not visibly soiled | 10% (5) | 0% (0) | 8% (2) | 14% (3) | 0% (0) |
| **Water requirements** | | | | | | |
| Water services | Running water | 43% (22) | 0% (0) | 16% (8) | 55% (12) | 40% (2) |
| | Piped water system connected to tap | 16% (8) | 0% (0) | 13% (3) | 18% (4) | 20% (1) |
| | Water trucking, storage or manual refilling | 14% (7) | 0% (0) | 13% (3) | 14% (3) | 20% (1) |
| Other water sources | Rainwater | 8% (4) | 0% (0) | 17% (4) | 0% (0) | 0% (0) |
| Wastewater management | Drainage system | 16% (8) | 0% (0) | 8% (2) | 23% (5) | 20% (1) |
| | Covered soakaway pit | 12% (6) | 0% (0) | 21% (5) | 0% (0) | 20% (1) |
| **Hand hygiene stations** | | | | | | |
| Handwashing station | Washbasin | 16% (8) | 0% (0) | 13% (3) | 18% (4) | 20% (1) |
| | Bucket or container connected to tap | 14% (7) | 50% (1) | 17% (4) | 5% (1) | 40% (2) |
| | Tippy tap | 12% (6) | 0% (0) | 4% (1) | 14% (3) | 40% (2) |

Continued

**Table 1** Continued

| Parameter | Recommendation | All guidelines (n=51) | Domestic setting (n=2) | Public setting (n=24) | Institutional setting (n=22) | Multiple settings (n=5) |
|---|---|---|---|---|---|---|
| Location of handwashing stations | Key entry/exit points | 27% (14) | 0% (0) | 50% (12) | 5% (1) | 20% (1) |
| | Close proximity to toilets | 27% (14) | 0% (0) | 21% (5) | 14% (7) | 40% (2) |
| | Food preparation and eating areas | 18% (9) | 0% (0) | 8% (2) | 27% (6) | 20% (1) |
| Accessibility | Accessible for all users and vulnerable groups | 12% (6) | 0% (0) | 13% (3) | 5% (1) | 40% (2) |
| | Height of soap and water taps appropriate for all | 10% (5) | 0% (0) | 13% (3) | 5% (1) | 20% (1) |
| | Age-appropriate handwashing stations | 8% (4) | 0% (0) | 0% (0) | 18% (4) | 0% (0) |
| Design considerations | Theft resistance | 16% (8) | 0% (0) | 17% (4) | 14% (3) | 20% (1) |
| | Water-saving designs/technologies | 12% (6) | 0% (0) | 13% (3) | 9% (2) | 20% (1) |
| COVID-19 adaptations | Taps that limit cross-contamination | 16% (8) | 0% (0) | 14% (7) | 0% (0) | 20% (1) |
| | 1–2 m spacing between handwashing stations | 12% (6) | 0% (0) | 17% (4) | 5% (1) | 20% (1) |
| | Disinfect taps regularly | 10% (5) | 0% (0) | 17% (4) | 0% (0) | 20% (1) |
| | Towels for opening and closing taps | 8% (4) | 0% (0) | 13% (3) | 5% (1) | 0% (0) |
| Availability of materials | Locally available materials for handwashing | 10% (5) | 0% (0) | 8% (2) | 9% (2) | 20% (1) |
| **Behaviour change** | | | | | | |
| Behaviour change techniques* | Prompts and cues | 12% (6) | 0% (0) | 13% (3) | 14% (3) | 0% (0) |
| | Habit formation | 8% (4) | 0% (0) | 13% (3) | 5% (1) | 0% (0) |
| Content of behaviour change messaging | Doable instructions with proper steps | 24% (12) | 0% (0) | 12% (6) | 27% (6) | 0% (0) |
| | Messages that target motivation | 10% (5) | 0% (0) | 8% (2) | 14% (3) | 0% (0) |
| Delivery channels | Visual reminders | 31% (16) | 0% (0) | 12% (6) | 36% (8) | 40% (2) |
| | Mass communication | 24% (12) | 0% (0) | 14% (7) | 9% (2) | 60% (3) |
| | Small group activities | 18% (9) | 0% (0) | 12% (6) | 9% (2) | 20% (1) |
| | Interpersonal communication | 10% (5) | 0% (0) | 17% (4) | 0% (0) | 20% (1) |
| Formative research | Identify determinants of target behaviour | 8% (4) | 0% (0) | 4% (1) | 0% (0) | 60% (3) |

*Using a typology of behaviour change techniques developed by Michie et al.[75]
ABHR, alcohol-based hand rub.

specify a quantitative standard nor whether non-drinking quality water may conditionally be used if high-quality water is not available. In addition, 10% of guidelines recommend that free residual chlorine must be greater than or equal to 0.5 mg/L after at least 30 min of contact time.[43 48 63 68 69]

### Alternative materials for hand hygiene

There are consistent recommendations for ABHR as an alternative material for hand hygiene. Of the guidelines that recommend ABHR, 35% (n=18) recommend ABHR with at least 60% alcohol, 22% (n=11) do not specify an alcohol percentage and only 6% (n=3) recommend an alcohol percentage of at least 70%. In addition, 18% of guidelines recommend ABHR where soap and water are not available[37–39 41 42 52–54 70] and 10% (n=5) only if hands are not visibly soiled.[21 36 40 65 71]

There are discordant recommendations for the use of ash. Twenty-two per cent of guidelines recommend ash as an alternative material to soap for handwashing.[12 36 43 45 47 52 64 65 71–73] In addition, 14% recommend ash if ABHR or soap is not available.[12 43 45 47 52 64 73] However, 6% (n=3) of guidelines advise against the use of ash or other products, such as soil, sand, mud or water alone.[45 47 62] In addition, while some guidelines (8%) recommend 0.05% chlorine solution,[48 53 54 63] one guideline advises against it.[13]

When disaggregated by setting, ABHR is most consistently recommended in public and institutional settings where it may meet larger and more frequent demand than handwashing stations. Similarly, recommendations for the conditional use of ABHR where soap and water are not available feature the most in the institutional setting, particularly in the workplace and schools. Ash

**Table 2** Parameters with gaps in recommendations (ie, fewer than 10 recommendations, equivalent to less than 20% of guidelines providing a recommendation)

| Parameters with gaps in recommendations* | Percentage of guidelines that provide a recommendation (n) |
|---|---|
| Effective hand hygiene | |
| Definition of hand hygiene | 4 (2) |
| Definition of safe/effective hand hygiene | 6 (3) |
| Handwashing knowledge | 16 (8) |
| Hand rubbing duration | 6 (3) |
| Minimum requirements for effective hand hygiene | |
| Soap and water requirements for handwashing | |
| Quantity of soap | 14 (7) |
| Quantity of water | 18 (9) |
| Hand hygiene stations | |
| Supply chain for products and materials | 6 (3) |
| Functionality | 10 (5) |
| Cost and affordability | 14 (7) |
| Durability | 16 (8) |
| Operation and maintenance—responsibility | 16 (8) |
| Operation and maintenance—actions | 18 (9) |
| Spacing and number of users per handwashing station | 18 (9) |
| Behaviour and behaviour change | |
| Frequency of behaviour change interventions | 4 (2) |
| Behaviour change approaches | 18 (9) |
| Government measures | |
| Sector policy and strategy | 0 (0) |
| Sector financing | 0 (0) |
| Capacity development | 2 (1) |
| Equity | 2 (1) |
| Planning, monitoring and review | 4 (2) |
| Institutional arrangements | 10 (5) |

*Parameters with less than 20% of guidelines that provide a recommendation.

is most commonly recommended in the public setting, which includes low-resource and water-scarce settings where soap and water may not be available.

### Hand hygiene stations

Overall, there are inconsistent recommendations on hand hygiene facilities and their location, as well as gaps. Across all settings, guidelines recommend washbasins (eg, ceramic, cement or plastic) (16%), a bucket or container connected to tap (14%) and tippy taps (12%) for hand-washing stations (table 1). Guidelines specify 16 different locations for handwashing stations, which include by the entrance and exit of public spaces and buildings (eg, restaurants, shops, markets, places of worship, train and bus stations) (27%), in close proximity to toilets (27%), and next to food preparation and eating areas (18%). Guidelines also mention placing hand hygiene stations, such as ABHR dispensers, at key entry and exit points of public spaces and buildings (14%). There are inconsistent recommendations on the optimal spacing and number of users per handwashing station, whereas none for ABHR dispensers. Gaps in recommendations include those for hand hygiene materials and product supply chains, cost and affordability, functionality, durability, and hand hygiene station operation and maintenance responsibilities and actions (table 2).

Recommendations on the location of handwashing stations vary slightly by setting. Close proximity to toilets is consistently recommended for both public and institutional settings (10% and 14%, respectively). By the entrance and exit of public spaces is most consistently recommended for public settings (24%). Next to food preparation and eating areas is most consistently recommended for institutional settings (12%), such as schools and the workplace.

### Hand hygiene station access and adaptations

Guidelines recommend hand hygiene stations that are accessible for all and adapted for pandemic response. For example, 10% of guidelines recommend that the height of soap and water taps be appropriate for access by children, the elderly and disabled (eg, 500–700 mm basin height for children and 850 mm basin height for wheelchair access) (table 1). Forty-five per cent (n=23) of guidelines recommend COVID-19-related adaptations for hand hygiene stations, such as taps that limit cross-contamination (16%), 1–2 m spacing between stations (12%), regular tap disinfection (10%), and towels for opening and closing taps (8%). Other adaptations include theft resistance (eg, attaching soap or other movable pieces to the station) (16%) and water-saving designs (eg, low-flow faucets) (12%).

### Recommendations for behaviour change

Overall, there are few recommendations related to hand hygiene behaviour change, though there are consistent recommendations on behaviour change messaging and delivery channels. For behaviour change messaging, some guidelines (10%) recommend messages that target 'motives'[74] (table 1).[20 45 54 64 71] In terms of delivery channels, 31% of guidelines recommend using visual reminders (eg, signs, posters or leaflets) and 24% mass communication (eg, radio, social media or mobile phone text messaging) to deliver behaviour change messages. Prompts, cues and habit formation are fairly consistently recommended as behaviour change techniques.[20 39 52 67 71 75–78] Formative research for behaviour change programmes

is recommended by 10% (n=5) of guidelines,[46 47 49 72 76] though only 8% (n=4) specifically mention undertaking formative research to identify behavioural determinants among the target population.[46 47 49 72] Guidelines provide inconsistent recommendations on behaviour change approaches (18%), determinants of hand hygiene to target for interventions (22%), and behaviour change models or frameworks (33%). Lastly, there are gaps in recommendations on frequency of behaviour change interventions (ie, how often an intervention should be delivered) and behaviour change approaches (4%)[20 46] (table 2).

### Recommendations for government measures

Overall, few guidelines (22%, n=11) provide a recommendation on government measures. Nonetheless, recommendations for sector policy and strategy include promoting local soap production and fostering public–private partnerships for handwashing (4%).[47 49] For institutional arrangements, 4% of guidelines suggest identifying ways of cross-sectoral collaboration for hand hygiene,[20 49] while other recommendations centre on engaging communities, the private sector and civil society for the delivery of WASH services.[12 65] On planning, monitoring and review, 4% of guidelines recommend supporting or reinforcing existing monitoring systems or creating a government-led national monitoring system, in line with global hygiene indicators.[49 68]

### Evidence-based recommendations

Of the 923 recommendations extracted from the 51 included guidelines, most (93%) do not have a citation for one of the five levels of evidence identified for the review (ie, systematic review, randomised and quasi-randomised trial, guideline, observational study

or anecdotal information) (figure 3). Of the remaining 7% of recommendations, less than 1% cite evidence from a systematic review.[46] In addition, less than 1% cite evidence from a randomised or quasi-randomised trial,[12 36] while over 1% cite evidence from observational studies, which are mainly recommendations for alternative hand hygiene materials, such as ash, sand and soil, or alternative water sources for handwashing, such as cooking or laundry water, bathwater and seawater.[12 36 47] Less than 1% of recommendations cite other guidelines developed from systematic reviews,[21 47 67 70] while 2.5% cite other guidelines not developed from systematic reviews,[51 56 57 65 71 78 79] such as WHO interim guidelines for hand hygiene in the context of COVID-19.[51 56 57 65 71 78 79] The other 2% of recommendations cite anecdotal information, such as programme documentation.[45 47 62 76]

### DISCUSSION

We identified 51 guidelines published between 1999 and 2021 by various international agencies covering a range of community settings. Most guidelines target the public and institutional settings, while surprisingly none exclusively target the domestic setting. Overall, community settings are not clearly defined among the guidelines, which presents an opportunity for future normative guidelines to establish a clear and common definition, especially as it relates to hand hygiene. Overall, no guidelines comprehensively address hand hygiene across domestic, public and institutional settings, and very few recommendations are evidence based, highlighting a gap in global normative guidance on hand hygiene in community settings.

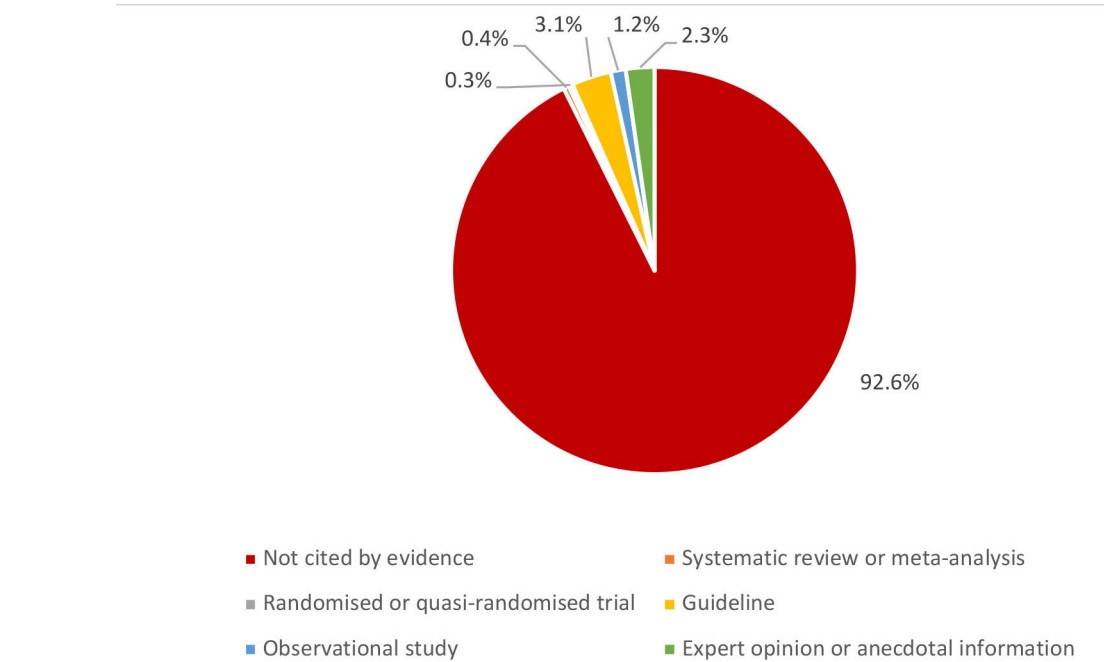

**Figure 3** Percentage of recommendations cited by one of the five types of evidence or no evidence.

## Effective hand hygiene

There is agreement among guidelines on what constitutes effective hand hygiene; however, inconsistencies and gaps remain as to when and how to practise hand hygiene. Current recommendations for handwashing with soap reflect findings from published literature, which show that handwashing with soap is an effective means for preventing the transmission of a range of diseases.[3–9] Regarding when to practise hand hygiene, future guidelines could focus key moments on those most likely to interrupt the transmission of infectious diseases in domestic, public and institutional settings. These might include 'after using the toilet' to reduce the transmission of diarrhoea-causing pathogens,[9] and 'after touching high-contact surfaces' or 'after coughing or sneezing' to reduce the risk of respiratory infections.[3] Lastly, gaps in recommendations on handwashing and hand rubbing technique present an opportunity for future guidelines to recommend the optimal technique for reducing bacterial load on hands.

## Minimum requirements

Consistent recommendations for the use of soap and water and ABHR suggest the widespread acceptability of these materials for effective hand hygiene across community settings. Current recommendations for the use of plain soap are consistent with findings from a systematic review that suggest that plain soap is more effective than antibacterial soap at removing or inactivating pathogens on hands in community settings.[80] Recommendations for running water are equally relevant as unreliable water supply negatively affects households' ability to perform hand hygiene.[81] Gaps in recommendations on minimum quantities of soap and water required for effective hand hygiene are also important for future guidelines to address. The recommended use of ABHR with at least 60% alcohol is in line with several studies which have found that ABHRs with an alcohol concentration between 60% and 80% are more effective at killing germs than those with a lower alcohol percentage, particularly in clinical settings.[82 83] Although ABHR can still inactivate many types of microbes when used correctly, evidence suggests that soap and running water are still more effective at removing certain types of pathogens that may be present on hands.[1] The WHO Guidelines on Hand Hygiene in Health Care, for example, recommend ABHR as the preferred method for routine hand hygiene in healthcare settings when hands are not visibly soiled, as it enables more frequent hand hygiene.[28] Similarly, in community settings, ABHR may be suitable in contexts where frequent hand hygiene is necessary, such as transport hubs and entrances or exits to public spaces and buildings.[84] ABHR may also be favourable where frequent hand hygiene is required, as the repeated use of soap can result in skin dryness and cause chronic irritant contact dermatitis.[28] However, in certain domestic, public and institutional settings, such as the household or schools, where hands may become more soiled, ABHR

may be less likely to effectively inactivate microbes.[82 83] As per current international guideline recommendations, handwashing with soap may therefore be prioritised in these settings, with ABHR as a suitable complement or alternative where frequent hand hygiene is required. Finally, because the transmission of germs is more likely to occur to and from wet hands, hand drying is an essential component of effective hand hygiene, especially for handwashing.[85] Current recommendations for hand drying in public and institutional settings are consistent with those in the WHO Guidelines on Hand Hygiene in Health Care, which recommend that hands should ideally be dried with individual paper towels, otherwise with air dryers.[28] While there is mixed evidence for the most effective hand drying method,[86] the WHO recommendations are based on findings that suggest that paper towels may effectively prevent recontamination of hands, while also lowering the risk of spreading pathogens through the air compared with electric air dryers.[85]

Discordant recommendations for minimum requirements suggest the need to leverage further research to determine the optimal water quality for handwashing and effectiveness of alternative materials for hand hygiene where soap and ABHR are not widely available. Limited evidence suggests that the use of non-potable water with low-to-moderate levels of *Escherichia coli* contamination may still be effective for handwashing,[87] which may be promising for areas where it is difficult to regularly treat water or where there is intermittent water supply that is prone to contamination. Similarly, two studies found that drinkable water may not be needed for handwashing with soap.[88 89] Nevertheless, the WHO Guidelines on Hand Hygiene in Health Care recommend washing hands with clean, running water whenever possible.[28] While there are discordant recommendations for the use of ash, there is uncertain evidence whether this stops or reduces the spread of pathogens compared with hand cleansing with soap, mud, soil or no hand cleansing.[90] Future guidelines may consider the efficacy of hand hygiene products along with their availability and acceptability in domestic, public and institutional settings to make relevant recommendations, particularly in water-scarce regions or settings where there is limited access to soap or ABHR. For example, soapy water may be a promising low-cost and effective alternative to bar soap in settings where bar or liquid soap is unaffordable.[91] In addition, one interim guideline on hand hygiene practices in low-resource settings, for example, recommends the use of friction-generating materials where clean, running water, soap or ABHR is not available.[36]

Current inconsistent recommendations for hand hygiene facilities and their placement may limit the practice of effective hand hygiene. Sustaining hand hygiene behaviour change requires consistent access to functional hand hygiene stations at key locations,[92 93] and diverse infrastructure is recommended with varying costs. Nonetheless, the appropriateness of these recommendations is likely to depend on the local availability and affordability

of materials. One guideline, for example, provides technical recommendations for permanent and semi-permanent handwashing facilities in public places and buildings, focusing on the sustainability and equitable access of these facilities.[20] Similarly, future guidelines may prioritise the accessibility, affordability, and sustainability of materials and infrastructure for hand hygiene across all community settings to address current inconsistent and discordant recommendations. The accessibility and sustainability of hand hygiene stations are particularly important to ensure that they are inclusive and kept functional and well stocked beyond their installation.[20]

## Behaviour change

The gaps in recommendations related to behaviour change suggest the need for guidance based on established behavioural theory and existing evidence. There are some recommendations for behaviour change, though without clear steps on how to develop, implement and sustain hand hygiene behaviour change interventions. Future guidelines may benefit from leveraging well-established behavioural frameworks and theories[94–96] to make recommendations as to how to develop effective, locally appropriate strategies beyond information-focused communication. In addition, while most behaviour change theories and frameworks recommended among the guidelines note the importance of formative research, very few guidelines recommend undertaking formative research. Yet, formative research plays a key role in adapting hand hygiene behaviour change programmes to high-risk populations and target settings.[97]

## Government measures

The significant gaps in recommendations on government measures underscore the current lack of normative standards to guide national governments on the planning, delivery, financing and monitoring of effective hand hygiene. Indicators also suggest inadequate planning and insufficient funding for hand hygiene among national governments globally.[19] Future guidelines may therefore consider prioritising government measures to support countries in responding to and preventing public health crises, such as the COVID-19 pandemic. Future guidelines may also focus recommendations on hand hygiene monitoring and reporting to improve comparison of hand hygiene indicators within and between countries.

## Strengths and limitations

This review has four main limitations. First, as a scoping review, we did not systematically assess the quality of the included guidelines, although we did, for example, consider aspects such as the extent to which recommendations were based on cited evidence. In addition, included guidelines covered WASH and public health topics beyond hand hygiene, so it was thus not necessarily relevant to assess the whole guidelines for quality, but rather focus on the robustness of the specific recommendations for hand hygiene. Second, the search was limited

to guidelines published in English or French and therefore may have excluded relevant guidelines published in other languages. Third, the lack of recommendations for government measures among current international guidelines may not reflect an absolute absence of guidance in this area. Guidance to governments on how to improve uptake of hand hygiene practices in community settings may be included in other sector documents or policy instruments. Fourth, while recommendations were summarised across domestic, public and institutional settings, there were often too few recommendations for each setting to assess consistency and concordance. Still, with 51 guidelines providing over 900 recommendations for hand hygiene in community settings, findings from this review highlight significant gaps and inconsistencies across community settings that future guidelines may seek to prioritise.

## CONCLUSION

This review identified 51 current international guidelines providing 923 recommendations for hand hygiene in community settings. Nonetheless, there are several important areas of discordance and significant gaps in the recommendations among these guidelines. Furthermore, very few recommendations are supported by any qualifying evidence. The COVID-19 pandemic led to numerous national, regional and international efforts to improve effective hand hygiene in domestic, public and institutional settings, such as households, public spaces, workplaces and schools, but the lack of clear recommendations supported by cited evidence may limit progress in this important area of public health.

**Author affiliations**
[1]Department of Disease Control, London School of Hygiene and Tropical Medicine Faculty of Infectious and Tropical Diseases, London, UK
[2]Hubert Department of Global Health, Rollins School of Public Health, Emory University, Atlanta, Georgia, USA
[3]Water and Sanitation Program, World Bank Group, Washington, District of Columbia, USA
[4]Department of Environmental Health and WASHTED, Malawi University of Business and Applied Sciences, Blantyre, Malawi
[5]Social and Behavioural Science Department, Center for Infectious Disease Research in Zambia, Lusaka, Zambia
[6]Department of Public Health, College of Medical Sciences, University of Calabar, Calabar, Nigeria
[7]Water, Sanitation, Hygiene and Health Unit, WHO, Geneva, Switzerland

**Contributors** CM, LB, BAC, CC, KC, JC, RD, RE-N, BG, JEM and OC informed the study protocol. CM carried out the database and grey literature search with input from OC and JEM. JEM, OC, BG and CM carried out expert consultations. CM and LB screened the retrieved articles for inclusion and extracted the data with input from OC. CM led the data analysis, while CM, LB, OC and JEM led the presentation of results with inputs from coauthors. CM, LB, JEM and OC led the writing of the manuscript with input from all coauthors. OC, JEM and BG provided overall supervision, leadership and advice. CM is the guarantor.

**Funding** This research was funded by the WHO and the UK Foreign, Commonwealth and Development Office.

**Disclaimer** The author is a staff member of the World Health Organization. The author alone is responsible for the views expressed in this publication and they

do not necessarily represent the views, decisions or policies of the World Health Organization.

**Competing interests** None declared.

**Patient and public involvement** Patients and/or the public were not involved in the design, or conduct, or reporting, or dissemination plans of this research.

**Patient consent for publication** Not required.

**Ethics approval** This research did not require institutional review board approval as the data were publicly available and collected from existing online databases and search engines. This research did not involve any human subjects.

**Provenance and peer review** Not commissioned; externally peer reviewed.

**Data availability statement** Data are available upon reasonable request.

**Open access** This is an open access article distributed under the terms of the Creative Commons Attribution IGO License (CC BY 3.0 IGO), which permits use, distribution,and reproduction in any medium, provided the original work is properly cited. In any reproduction of this article there should not be any suggestion that WHO or this article endorse any specific organization or products. The use of the WHO logo is not permitted. This notice should be preserved along with the article's original URL.

**ORCID iDs**
Clara MacLeod http://orcid.org/0000-0001-6952-9103
Oliver Cumming http://orcid.org/0000-0002-5074-8709

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
