## [Reviewer comments · BMJ Open]

ARTICLE DETAILS

TITLE (PROVISIONAL)	Recommendations for hand hygiene in community settings: a scoping review of current international guidelines
AUTHORS	MacLeod, Clara; Braun, Laura; Caruso, Bethany; Chase, Claire; Chidziwisano, Kondwani; Chipungu, Jenala; Dreibelbis, Robert; Ejemot-Nwadiaro, Regina; Gordon, Bruce; Esteves Mills, Joanna; Cumming, Oliver

VERSION 1 – REVIEW

REVIEWER	Müller, Sophie Robert Koch Institut
REVIEW RETURNED	11-Oct-2022

GENERAL COMMENTS	Congratulations to the overall well written scoping review on “Improving hand hygiene in community settings: a scoping review of current international guidelines” Please see below my detailed points for your consideration Methods: - Please further elaborate why you opted for including nursing homes. In the general opinion, if it is not nursing homes run by relatives and families, nursing homes belong to the healthcare settings as professionals or at least staff with a training work here- Consider adding “strategy” or “standard” to your search term, does this give you more results?- Search in Google: Google only showed 32 results? Or did you only include the first pages of the google output?- Kindly share the data extraction template as annex Results - Overall references in the written part are missing. It is not possible for the reader to deduct from the text or table which included guideline is meant. Please make sure to add the references. The reader might for example be interested in knowing which guideline recommends ash.- Page 7: “31 providing COVID_19-specific guidance”. This needs further information. Please further explain and add information in abstract and discussion section. Are those guidelines generalizable or do they only apply in pandemic times?- Page 8 line 35: “including”? take out?- Page 9: Handwashing requirements. This section needs further detail. Do the guidelines really recommend singles use paper towels at home? Following Table 1 this is not the case. But info is not clear at the first sight. Is there really no data at all on “hand drying” for the domestic setting?
---

	 - Please also give more detail on the minimal requirements at different settings in LMIC and HICs, as this info is needed for policy and implementation guidelines. - Table 1 numbers don't add up in "before and after preparing food", "soap" - Table 1 %missing in "Location of household stations". - Table 2: the "no" in "no definition" should be taken out or consistently all lines would need a "no". - The concept "quantity of soap" appears hard to grasp, kindly give an example for bar soap. - Page 14: line 32-43 are redundant, please rephrase - Personally I find the lack in evidence based recommendations shocking. Please give more detail, do recommendations don't have any kind of evidence, where do they come from e.g. expert opinion? - Page 15 line 52, add "" in "after using..", "after touching..", "after coughing". Discussion  - Page 16 line 25: elaborate on skin dryness, as a potential reason for the recommendation of ABHR when frequent hand hygiene is needed - Page 16, line 35: please add a remark for domestic settings mirroring that no guideline actually recommends "paper towels" in domestic settings Supplement:  - Table A7: critical control point language, not clear, please clarify
--	--

REVIEWER	McDonald , Margaret V Center for Home Care Policy & Research, Visiting Nurse Service of New York
REVIEW RETURNED	01-Nov-2022

GENERAL COMMENTS	The authors sought to review and evaluate international guidelines on hand hygiene practices that would reduce transmission of pathogens in community settings. The team was comprehensive in describing their search terms and review processes. They broke down the recommendations into four areas: i) effective hand hygiene, ii) minimal requirements, iii) behaviour change, iv) government measures. They were successful in highlighting the preferred and consistent recommendations of use of soap and water with alcohol-based hand rub (ABHR) as an alternative of complement; and highlighting length of time for washing processes. They point out important inconsistencies/gaps in the recommendations, in particular about water quality, soap and water quantity, and effective alternatives to soap and alcohol-based hand rub (ABHR). While authors also synthesized information on how to improve uptake of recommendations including which government measures would be required for effective hand hygiene, these are elements that are not always/usually found in guidelines so important information relating to these areas may be missing and should be noted in the limitations. Also, the authors highlight that less than 10% of recommendation are support by evidence but it is really that less than 10% indicate where the recommendations come form – so lack of citation of the evidence, rather than them not being evidence-based. The methods section does indicate how evidence based is defined for the article purpose "A recommendation was considered evidence-based if the guideline provided a specific citation for the
--

	recommendation” but the statement that they are not evidence based is too strong. The tables are very useful and easy to follow. Overall, a very comprehensive scoping review that contributes to the conversation about hand hygiene; laying the framework on how to improve guidelines. A bit more clarity on how the guidelines conflict versus where the guideline gaps are would be helpful. Specific recommendations: Title: “Improving hand hygiene in community settings” leads one to think that it is a scoping review for behaviour change. While one of the four areas that the authors ultimately mapped to including behaviour change, the scoping review primarily focuses on reviewing recommendations of best practices for hand hygiene in community settings so would suggest changing the title and text to reflect this focus. Abstract: Results – “Overall, less than 10% of recommendations are supported by evidence” would change to something like “Overall, less than 10% noted if the recommendations were evidence-based.” Conclusions: Seems a bit strong to suggest that there is a lack of consistent, evidence-based recommendations and that it may constrain global efforts. There are some importance consistencies but there are gaps that still need to be addressed – would focus on this in the conclusion. Results: In some of the sections it is unclear when there is an inconsistency versus when there is a gap. Clear example: Alternative materials subsection: 22% of guidelines recommend ash as an alternative material....14% of guidelines advise against the use of ash.” Less clear example: Hand hygiene subsection – conflicting recommendations or just different number of mentions in the guidelines? Minor edit: In Strengths and limitation of this study – the word “in” is missing in first bullet “...for hand hygiene community settings”
--	---

VERSION 1 – AUTHOR RESPONSE

Reviewer 1		
Methods		
Please further elaborate why you opted for including nursing homes. In the general opinion, if it is not nursing homes run by relatives and families, nursing homes belong to the healthcare settings as professionals or at least staff with a training work here	Thank you, this is an important point which our team discussed at length in preparing the protocol. We decided to include nursing homes in the scoping review, as guidelines for hand hygiene in nursing homes often include recommendations for hand hygiene in non-acute care	“Nursing homes, long-term care facilities, non-acute care facilities, and home care are also included in the review under institutional settings, as the boundary between healthcare and non-healthcare settings is often unclear or recommendations for these sub-settings may not be disaggregated, especially in the context of COVID-19 where people received care at home.” (page 6, lines 20-24)

	facilities and home care. As the boundary between healthcare and non-healthcare is often not clear in these settings, we opted to include these recommendations rather than exclude as they may have relevance to policy and practice outside of healthcare settings. An example of this is the recent experience during the COVID-19 pandemic where older people often received care at home and where hand hygiene was part of the measures to prevent transmission.	
Consider adding “strategy” or “standard” to your search term, does this give you more results?	Thank you for the suggestion, which we acted upon. To assess whether the exclusion of these terms limited our results, we re-ran our searches adding both terms. Adding these terms did not produce any additional results in the WHO IRIS database nor in the Google search function.	n/a
Search in Google: Google only showed 32 results? Or did you only include the first pages of the google output?	The Google search was carried out by one reviewer that screened the first 10 pages of results. Documents that at first appeared related to the research question and met the inclusion criteria were included for further screening. More detail related to the Google engine search has been added to the “identifying relevant studies (stage 2)” paragraph.	“... The search in Google was carried out using the anonymous function in the web browser (Chrome) to reduce the influence of the reviewer’s (CM) individual search history. Search strings were constructed by using multiple combinations of search terms. The first 10 pages of Google were reviewed by one reviewer (CM). Documents that at first appeared related to the research question and met the inclusion criteria were included for further screening.” (pages 6, lines 39-45)
Kindly share the data extraction template as annex	Thank you – the data extraction template is now in the supplementary materials document – Table A1 – and	“The data extraction form (Table A5) included information on guideline characteristics, such as author, year of publication, target setting, and COVID-19 response, as well as 57 specific

	cross-referenced in the main document.	parameters related to the four areas of hand hygiene described in the conceptual framework” (page 7, line 20)
Results		
Overall references in the written part are missing. It is not possible for the reader to deduct from the text or table which included guideline is meant. Please make sure to add the references. The reader might for example be interested in knowing which guideline recommends ash.	Thank you for pointing this out. We have added guideline references for the recommendations throughout the results section where possible.	n/a
Page 7: “31 providing COVID_19-specific guidance”. This needs further information. Please further explain and add information in abstract and discussion section. Are those guidelines generalizable or do they only apply in pandemic times?	Thank you, this has been elaborated in the paragraph and in the abstract.	“The 51 included guidelines were published in English between 1999 and 2021, with 38 published in 2020 or later and 31 providing guidance specifically focusing on hand hygiene to help prevent the transmission of the COVID-19 virus. ” (page 8, lines 12-14)
Page 8 line 35: “including”? take out?	“Including” has been replaced with “providing” to make this clearer.	“Meanwhile, 24% of guidelines recommend the inclusion or provision of simple instructions on handwashing technique in hygiene promotion programmes.” (page 8, lines 51-52)
Page 9: Handwashing requirements. This section needs further detail. Do the guidelines really recommend singles use paper towels at home? Following Table 1 this is not the case. But info is not clear at the first sight. Is there really no data at all on “hand drying” for the domestic setting?	That is correct, there is no detail on hand drying for the domestic setting. We have added more detail though by disaggregating findings by setting (e.g., domestic, institutional, and public).	“Guidelines consistently recommend plain soap for handwashing across all settings and paper towels or electric dryers for hand drying in public and institutional settings. ” (page 9, lines 19-21) “For hand drying after handwashing, most guidelines recommend clean, single use paper towels (39%) and electric air-drying systems (22%) in public and institutional settings. 8% of guidelines recommend a bin with disposal liners and lid for the waste management of paper towels. [21,55–57] No guidelines provide

		recommendations for hand drying materials in the domestic setting.” (page 9, lines 24-29)
Please also give more detail on the minimal requirements at different settings in LMIC and HICs, as this info is needed for policy and implementation guidelines.	Guidelines did not explicitly distinguish recommendations for high-income and low-middle-income country settings, although some included guidelines focused on community settings in low-resource settings only. The synthesis of recommendations included in the scoping review are intended to be applicable to high-income and low-income countries, although specific considerations for low-resource settings or climate change adaptations are highlighted where possible.	“The majority of recommendations are generalisable to high-, medium-, and low-income settings.” (page 8, lines 25-26)
Table 1 numbers don’t add up in “before and after preparing food”, “soap”	Thank you for alerting us to this error. We have now revised the percentage and number for institutional settings.	Table 1 –  • Before and after preparing food, institutional settings: “18% (4)” (page 10, line 12) • Soap, all settings: “20% (1)” (page 10, line 25)
Table 1 %missing in “Location of household stations”.	We have added the percentages.	Table 1 – location of handwashing stations, all guidelines (page 11, lines 19-21):  • Key entry/exit points: 27% (14) • Close proximity to toilets: 27% (14) • Food preparation and eating areas: 18% (9)
Table 2: the “no” in “no definition” should be taken out or consistently all lines would need a “no”.	We have removed the “no’s” for both the definition of hand hygiene and safe hand hygiene. We have also added “effective” after “safe”, as guidelines use the terms interchangeably.	No definition of hand hygiene No definition of safe/effective hand hygiene” (page 13, lines 10-11)
The concept “quantity of soap” appears hard to grasp, kindly give an example for bar soap.	We have added more detail on the quantity of soap with specific examples for quantities of bar and liquid soap recommended among four guidelines.	“Few guidelines recommend a minimum quantity of liquid or bar soap (14%)[43,48,58–61] or water (18%)[12,20,47,50,60–62] required for handwashing (e.g., minimum amount of water needed to wet hands or soap product necessary to cover all surfaces of hands for handwashing with soap) (Table 2). One guideline, for example, recommends less than 5 ml of liquid soap per person per handwashing

		event,[60] whereas three other guidelines recommend at least 250 grams of bar soap per person per month for handwashing.[43,48,61]" (page 9, lines 37-44)
Page 14: line 32-43 are redundant, please rephrase	The paragraph has been revised to remove this redundancy.	"Guidelines recommend hand hygiene stations that are accessible for all, including people with disabilities and older adults, and adapted for pandemic response. For example, 12% of guidelines recommend that hand hygiene stations be accessible for all users and vulnerable groups. In addition, 10% of guidelines recommend that the height of soap and water taps be appropriate for access by children, the elderly, and disabled ..." (page 14, lines 34-37)
Personally I find the lack in evidence based recommendations shocking. Please give more detail, do recommendations don't have any kind of evidence, where do they come from, e.g., expert opinion?	We agree that this is an important and surprising result of our review. Recommendations were considered "evidence-based" if any evidence was cited for each recommendation. To clarify this, we adapted a framework to classify evidence cited for recommendations into five levels – we have added further detail on this in the methods section and have expanded on this in the results as well. Among the 10% of recommendations which did cite evidence, only a minority cited a systematic review or peer-reviewed literature.	Abstract: "Overall, less than 10% of recommendations were supported by any cited evidence." (page 3, line 28) Results: "Over 90% of recommendations do not have a citation for one of the five levels of evidence identified for the review (Figure 3). Of the 923 extracted recommendations, only 7% (n= 68) indicate whether the recommendation is evidence-based (i.e., cited by one of the five levels of evidence, i.e., systematic review, randomised and quasi-randomised trial, other guideline, observational study, or expert opinion or anecdotal information). Less than 1% (n= 4) of recommendations cite evidence from a systematic review.[46] In addition, less than 1% (n= 3) cite evidence from a randomised or quasi-randomised trial,[12,36] while over 1% (n= 11) cite evidence from observational studies, which are mainly recommendations for alternative hand hygiene materials, such as ash, sand, and soil, or alternative water sources for handwashing, such as cooking water, laundry water, bathwater, and seawater.[12,36,47] Less than 1% (n= 6) of recommendations cite other guidelines developed from systematic reviews,[21,47,67,71] while 2.5% (n= 23) cite WHO interim guidelines for hand hygiene in the context of COVID-19.[51,56,57,65,72,78,79] The other 2% (n= 21) of recommendations cite

		anecdotal information, such as programme documentation.[45,47,62,76] The remaining recommendations (93%, n= 855) lack any citation.” (page 15, lines 23-38) Conclusion: “... the lack of clear recommendations supported by cited evidence may limit progress in this important area of public health.” (page 18, line 28)
Page 15 line 52, add “” in “after using..”, “after touching..”, “after coughing”	We have added quotation marks for each key moment.	“These might include “after using the toilet” to reduce the transmission of diarrhoea-causing pathogens, and “after touching high-contact surfaces” or “after coughing or sneezing” to reduce the risk of respiratory infections.” (page 16, lines 4-7)
Discussion		
Page 16 line 25: elaborate on skin dryness, as a potential reason for the recommendation of ABHR when frequent hand hygiene is needed	This is a good point, thank you. We have added a sentence in the discussion to address this.	“ABHR may also be favourable where frequent hand hygiene is required, as the repeated use of soap can result in skin dryness and cause chronic irritant contact dermatitis.[28]” (page 16, lines 31-33)
Page 16, line 35: please add a remark for domestic settings mirroring that no guideline actually recommends “paper towels” in domestic settings	Thank you – we have clarified this by highlighting that the recommendations for paper towels and electric dryers are for the public and institutional settings.	“Current recommendations for hand drying in public and institutional settings are consistent with those in the WHO Guidelines on Hand Hygiene in Health Care ...” (page 16, lines 41-42)
Supplement		
Table A7: critical control point language, not clear, please clarify	This has been revised for clarity to also reflect the text included in the manuscript.	“Terminology for defining when to practice safe hand hygiene” (Supplemental Materials, Table A8)
Reviewer 2		
General		
“Improving hand hygiene in community settings” leads one to think that it is a scoping review for behaviour change. While one of the four areas that the authors ultimately mapped to including behaviour change, the	Thank you – we have revised the title to: “Recommendations for hand hygiene in community settings: a scoping review of current international guidelines”.	“Recommendations for hand hygiene in community settings: a scoping review of current international guidelines” (page 2, lines 5-7)

scoping review primarily focuses on reviewing recommendations of best practices for hand hygiene in community settings so would suggest changing the title and text to reflect this focus.		
---	--	--

Abstract

Results – “Overall, less than 10% of recommendations are supported by evidence” would change to something like “Overall, less than 10% noted if the recommendations were evidence-based.”	Please also refer to our response to Reviewer 1 on this where we have edited for clarity. We have revised our methodology for assessing evidence-based recommendations. We adapted a framework to classify evidence cited for recommendations into five levels, which are further explained in the methods section. We therefore only considered recommendations evidence-based if they cited one of the five levels of evidence. We also revised the results based on these levels of evidence accordingly, as well as updated Figure 3.	Abstract: “Overall, less than 10% of recommendations were supported by any cited evidence.” (page 3, line 28) Methods: “... We also used a hierarchical system to classify evidence cited for recommendations into five levels, adapted according to an established classification system: 1) systematic review or meta-analysis, 2) randomised controlled trial or controlled trial without randomisation (e.g., quasi-experimental), 3) guidelines developed from systematic reviews, 4) observational studies (e.g., cohort, cross-sectional, case-control studies), 5) expert opinion or anecdotal information (e.g., programme documentation). A recommendation was considered evidence-based if the guideline provided a specific citation for the recommendation, which was coded according to the level of evidence cited.” (page 7, lines 36-44) Results: Results: “Over 90% of recommendations do not have a citation for one of the five levels of evidence identified for the review (Figure 3). Of the 923 extracted recommendations, only 7% (n= 68) indicate whether the recommendation is evidence-based (i.e., cited by one of the five levels of evidence, i.e., systematic review, randomised and quasi-randomised trial, other guideline, observational study, or expert opinion or anecdotal information). Less than 1% (n= 4) of recommendations cite evidence from a systematic review.[46] In addition, less than 1% (n= 3) cite evidence from a randomised or quasi-randomised trial,[12,36] while over 1% (n= 11) cite evidence from observational studies, which are mainly recommendations for
--	---	--

		alternative hand hygiene materials, such as ash, sand, and soil, or alternative water sources for handwashing, such as cooking water, laundry water, bathwater, and seawater.[12,36,47] Less than 1% (n= 6) of recommendations cite other guidelines developed from systematic reviews,[21,47,67,71] while 2.5% (n= 23) cite WHO interim guidelines for hand hygiene in the context of COVID-19.[51,56,57,65,72,78,79] The other 2% (n= 21) of recommendations cite anecdotal information, such as programme documentation.[45,47,62,76] The remaining recommendations (93%, n= 855) lack any citation.” (page 15, lines 23-38)
Conclusions: Seems a bit strong to suggest that there is a lack of consistent, evidence-based recommendations and that it may constrain global efforts. There are some importance consistencies but there are gaps that still need to be addressed – would focus on this in the conclusion.	Thank you, this has been revised accordingly.	“While current international guidelines consistently recommend handwashing with soap in domestic, public, and institutional settings, there remain gaps where clear evidence-based guidance might support more effective policy and investment.” (page 3, lines 31-33)
Results		
In some of the sections it is unclear when there is an inconsistency versus when there is a gap. Clear example: Alternative materials subsection: 22% of guidelines recommend ash as an alternative material....14% of guidelines advise against the use of ash.” Less clear example: Hand hygiene subsection – conflicting recommendations or just different number of mentions in the guidelines?	We have addressed this distinction (inconsistency vs. gap) more explicitly where possible in the Results section, particularly in the hand hygiene sub-sections that you have highlighted.	“Guidelines use inconsistent terms for defining when to practice hand hygiene, such as ‘key times’ (14%), etc.” (page 8, lines 36-38) “There is a lack of recommendations on handwashing or hand rubbing technique (14%).” (page 8, lines 49-51) “Guidelines provide inconsistent recommendations on when to practice hand hygiene. Over 30 different individual key moments are recommended among the guidelines. Despite this inconsistency, the individual key moments most commonly

		mentioned among the guidelines include ...” (page 8, line 57 – page 9, line 4)
Minor edit		
In Strengths and limitation of this study – the word “in” is missing in first bullet “...for hand hygiene community settings”	We have added “in” between hand hygiene and community settings.	“This is the first scoping review to synthesise current recommendations for hand hygiene in community settings among international guidelines and assess to what extent they are consistent and evidence-based.” (page 4, line 5)

VERSION 2 – REVIEW

REVIEWER	Müller, Sophie Robert Koch Institut
REVIEW RETURNED	12-Dec-2022

GENERAL COMMENTS	Dear researchers, thank you very much for considering my comments. Please allow me to still have 2 comments that are not yet completely addressed. 1) "Search in Google: Google only showed 32 results? Or did you only include the first pages of the google output?" Please allow me to get back to this. You state in results “32 from google”. Did the first 10 pages of google show you 32 records? If I google, it always shows me 10 records per page, so I wonder how you got this number of 32 when screening the first 10 pages. 2) Less than 1% (n= 4) of recommendations cite evidence from a systematic review.[46] In addition, less than 1% (n= 3) cite evidence from a randomised or quasi-randomised trial,[12,36] while over 1% (n= 11) cite evidence from observational studies, which are mainly recommendations for alternative hand hygiene materials, such as ash, sand, and soil, or alternative water sources for handwashing, such as cooking water, laundry water, bathwater, and seawater.[12,36,47] Less than 1% (n= 6) of recommendations cite other guidelines developed from systematic reviews,[21,47,67,71] Please explain why the number n does not match the references supporting this information. This is the case for N=4, N=3 and N=6 in above statements
--

REVIEWER	McDonald , Margaret V Center for Home Care Policy & Research, Visiting Nurse Service of New York
REVIEW RETURNED	16-Dec-2022

GENERAL COMMENTS	Revision suggestions have been addressed.
------------------	---

VERSION 2 – AUTHOR RESPONSE

Reviewer 1		
Methods		
Please allow me to get back to this. You state in results “32 from google”. Did the first 10 pages of google show you 32 records? If I google, it always shows me 10 records per page, so I wonder how you got this number of 32 when screening the first 10 pages.	Thank you – the PRISMA flow diagram, as well as the corresponding text in the results section, has been revised to reflect this. The number of documents identified via Google has been changed from 32 to 900 to reflect the number of results from the Google search using the different search term combinations. The 32, which refers to the number of documents sought for retrieval from Google, is included in the total number of documents sought for retrieval (n= 125).	 • Figure 2 – PRISMA flow diagram • Search results: “Electronic searches were conducted on 15 November 2021, identifying 3,360 records (2,432 from the WHO IRIS database, 900 from Google, 28 from international agency websites) (Table A2) and a further 11 records identified through expert consultation. Following de-duplication, a total of 3,132 records were screened by title and abstract” (page 7, line 57 – page 8, line 4)
Please explain why the number n does not match the references supporting this information. This is the case for N=4, N=3 and N=6 in above statements.	Thank you for pointing this out. The paragraph has been revised for clarity. In some instances, the number does not match the number of references as guidelines may provide more than one recommendation cited by one of the five levels of evidence. For n=4, for example, the 4 recommendations supported by results from a systematic review are extracted from the same guideline, which is included as the reference.	“Of the 923 recommendations extracted from the 51 included guidelines, most (93%) do not have a citation for one of the five levels of evidence identified for the review (i.e., systematic review, randomised and quasi-randomised trial, guideline, observational study, or anecdotal information) (Figure 3). Of the remaining 7% of recommendations, less than 1% cite evidence from a systematic review.[46] In addition, less than 1% cite evidence from a randomised or quasi-randomised trial,[12,36] while over 1% cite evidence from observational studies, which are mainly recommendations for alternative hand hygiene materials, such as ash, sand, and soil, or alternative water sources for handwashing, such as cooking water, laundry water, bathwater, and seawater.[12,36,47] Less than 1% of recommendations cite other guidelines

		developed from systematic reviews,[21,47,67,71] while 2.5% cite guidelines not developed from systematic reviews, such as WHO interim guidelines for hand hygiene in the context of COVID-19.[51,56,57,65,72,78,79] The other 2% of recommendations cite anecdotal information, such as programme documentation.[45,47,62,76]" (page 15, lines 23 – 47)
--	--	---